# Prompting Pedagogical Change through Promoting Active Lifestyles Paradoxes

**DOI:** 10.3390/ijerph17217965

**Published:** 2020-10-29

**Authors:** Jo Harris, Lorraine Cale, Oliver Hooper

**Affiliations:** School of Sport, Exercise and Health Sciences, Loughborough University, Loughborough LE11 3TU, UK; l.a.cale@lboro.ac.uk (L.C.); o.r.hooper@lboro.ac.uk (O.H.)

**Keywords:** health promotion, physical activity promotion, pedagogy, professional development

## Abstract

This study explored approaches to promoting active lifestyles (PAL) in schools which aimed to inform and develop physical education (PE) trainees and teachers’ health-related philosophies and pedagogies. Thirty-two secondary school PE trainees and teachers involved in a University-based Initial Teacher Education partnership in England participated in this study during the period 2015–2019. The participants were involved in professional development which included an introduction to research-informed PAL ‘paradoxes’ associated with promoting active lifestyles. Participants were asked to review their own health-related philosophies and practices in light of these paradoxes and were encouraged to use them to influence their own pedagogies as well as those of colleagues/peers. Participants found the PAL paradoxes interesting, surprising and perplexing; expressed a keen desire to address and solve them; and experienced the joys and challenges of influencing colleagues’/peers’ health-related philosophies and pedagogies. The findings suggest that this innovative low-cost, flexible and accessible approach to pedagogical change has the potential to engage PE teachers, increase their effectiveness as promoters of physical activity, and to greatly enhance the subject’s contribution to public health. This is significant, given calls for new pedagogical approaches and teachers’ previously reported lack of engagement in professional development in this area.

## 1. Introduction

It is widely advocated that schools can and should play a major role in contributing to public health [1]. However, this can create tension between education and health outcomes as the priority of schools is clearly education and they have limited capacity and funding to take on additional public health outcomes [2]. Nonetheless, there is much logic in schools striving to effectively contribute to health promotion, given their skilled workforce and their ability to reach and influence virtually all children and their families. Consequently, schools have been encouraged to adopt whole-school approaches to health promotion and the pursuit of healthy behaviours, such as eating a healthy diet and being routinely active [3,4].

Furthermore, health is increasingly being recognised as the leading justification for PE in schools [5], with the promotion of active lifestyles an established goal of PE curricula across the world [6,7]. However, the appropriateness of this goal for the subject has been questioned within the PE profession. For example, O’Sullivan has outlined potential pitfalls of a public health agenda for PE and has encouraged the profession to find a sensible balance between the health expectations, the joy of being active, and the educational value of physical activity [8]. In addition, concerns have been expressed about the inconsistent expression of health within PE, including potentially negative practices such as an overemphasis on training regimes and fitness [9,10]. The above has led to calls for evidence-based PE-for-health pedagogies [11,12]. Indeed, this is rapidly becoming a developing area of research, with some encouraging outcomes reported to date. For example, positive effects were found by Weaver and colleagues [13,14] in the United States from PE teachers’ use of physical activity promoting practices (e.g., small-sided games; avoiding elimination activities) and by Bowler [15] and Sammon [16] in the United Kingdom on PE teachers’ practices following application of a pedagogical model for health-based physical education [17].

Given the expectations on schools in general and PE in particular to effectively promote active lifestyles, alongside concerns about current ways of approaching this, the PAL project was devised. The project aimed to explore approaches to promoting active lifestyles which were designed to inform and develop PE trainees’ and teachers’ health-related philosophies and pedagogies. It involved a short professional development course designed by University staff. This initially led to the co-development and implementation of whole-school and PE-specific principles associated with promoting active lifestyles which have been reported elsewhere [18,19], and later to the co-development of a series of paradoxes associated with promoting active lifestyles, an example being that PE lessons offer young people regular opportunities to be active yet activity levels within PE lessons are reportedly low. This particular paper focuses on the PE trainees’ and teachers’ engagement with the paradoxes.

## 2. Materials and Methods

From a theoretical perspective, the PAL project was underpinned by social cognitive theory which recognises the influence of experiences, actions and environmental factors on health behaviour [20] (Bandura, 1986). Similar to other studies focusing on the promotion of physical activity in school settings [21,22], it also embraced the social ecological model which acknowledges the interactive characteristics of individuals and environments that underlie health outcomes [23] and emphasises that children’s physical activity behaviour is influenced by multiple levels associated with individual, social, environmental and policy factors (Salmon and King) [24]. The PAL project focused in particular on the intrapersonal and interpersonal levels of the social ecological model [25] by providing support to PE trainees and teachers (and, in turn, their colleagues and peers) who regularly interact with and potentially influence young people’s health-related understanding, attitudes and behaviours.

The project was further informed by reviews on school-based approaches to health promotion. For example, Kriemler and colleagues’ review [26] confirmed the public health potential of high-quality, school-based physical activity interventions for increasing physical activity in youth. In addition, Golden and Earp’s review [27] revealed that interventions that focused on certain topics or occurred in particular settings more successfully implemented a social ecological approach. Furthermore, a review of interventions focusing specifically on increasing physical activity in the school setting found multi-faceted interventions to be the most effective, such as those which aimed to increase physical activity levels in PE lessons as well as incorporate curriculum and/or environmental changes; and strategies requiring little training and promoting less structured types of physical activity were also found to be more sustainable [28]. The findings of these reviews informed the design of the PAL project in that it adopted a single topic (physical activity), single setting (schools), multi-faceted approach (whole school and subject specific). It was also designed to be sufficiently flexible to be implemented across a range of school settings and populations, and to be sustainable in that it involved limited training and resources and there was no requirement for schools/teachers to follow a structured programme.

### 2.1. Participants

Secondary school PE teachers and trainee teachers involved in a University-based Initial Teacher Education (ITE) partnership in the East Midlands in England were invited to participate in this study. At the beginning of three consecutive academic years commencing in the period 2015–2016, the PE trainee teachers at the University and PE mentors (teachers with a minimum of three years’ teaching experience who support the trainee teachers’ development in the secondary schools within the partnership) were invited to be involved in the project. During the periods 2016–2017 and 2017–2018, the invitation was also sent to participants from previous years. In the fourth year of the project, no invitations were made to trainee teachers, but previous participants were invited to continue their involvement. Throughout the duration of this study, eight participants from the period 2015–2016, seven from the period 2016–2017 and seven from the period 2017–2018 remained involved, while 10 withdrew due to personal and health issues and/or increased work responsibilities. Of the 22 participants who engaged with this study throughout, 13 were female and 9 male. A summary of the participants is presented in Table 1.

### 2.2. Project Delivery

In the first year of the project, participants attended four, two-hour professional development sessions at the University which were scheduled every few months and facilitated by the authors. During the following three academic years, these sessions were replaced by a three hour professional development workshop again held at the University. During the University-based professional development, participants were informed about and discussed the implications of worldwide literature associated with the promotion of active lifestyles (e.g., that relating to whole-school approaches, physical activity for health recommendations, health-based PE, and the role of fitness testing in PE and in promoting activity). From the period 2016–2017, the participants were also introduced to a number of contradictions or paradoxes associated with promoting active lifestyles such as:The promotion of active lifestyles is usually prominent in teachers’ philosophies of PE, yet it is much less evident in PE curricula. For example, whilst PE teachers often claim to encourage and educate about long-term engagement in physical activity, their written schemes of work do not necessarily reflect this.PE lessons offer regular opportunities to be active, yet activity levels in PE are generally low. For example, whilst pupils may have regular PE lessons, it seems that they are not particularly active during PE lessons.PE teachers often claim to use fitness testing to promote activity, yet many pupils dislike and learn little from fitness testing. For example, fitness testing is frequently incorporated into PE curricula to encourage pupils to be active, but many do not enjoy the experience and gain limited knowledge and understanding from it.PE teachers help develop pupils’ knowledge and understanding about leading active lifestyles, yet many pupils are confused about this. For example, pupils have opportunities to gain good understanding about physical activity from PE lessons, but many have misconceptions associated with this such as thin people are healthy and you need to run fast to be healthy.

Each paradox was presented in the form of a resource which included a statement about the paradoxical issue plus a concise summary of the literature associated with the contrasting perspectives on the issue, followed by reflective questions about what might be done to address the paradox. Participants were asked to reflect on the PAL paradoxes and, if possible, think of additional paradoxes. During the project, participants identified the following additional paradox associated with the promotion of active lifestyles:

Technology can help motivate pupils to be active but it can also reduce activity levels (2017). For example, the use of activity trackers in PE lessons can encourage some pupils to be more active yet the use of tablets for demonstration and analysis of actions can result in less activity.

Participants were asked to review their own health-related philosophies and practices in light of these PAL paradoxes, and they were encouraged to use them to inform their own pedagogies as well as try and influence the health-related philosophies and pedagogies of their colleagues/peers. Participants were prompted to share their experiences via participant-led online groups. The groups were proposed as a means of establishing supportive communities of practice [29]. The authors did not engage with these groups although they did respond to communication from them, such as to requests for laminated copies of the PAL paradoxes to use in professional development sessions with PE colleagues and teachers in their respective schools.

### 2.3. Data Gathering and Analysis

Participants were invited to complete an online survey three months into and at the end of each academic year. The early survey included questions about the PE trainees’ and teachers’ views on and responses to the PAL paradoxes and any plans to use them, for example, in their own pedagogies and/or to influence colleagues’/peers’ health-related philosophies and pedagogies. The survey at the end of each academic year then included questions about the effects, if any, the PAL paradoxes had had on their own and colleagues’/peers’ health-related philosophies and pedagogies.

All participants were also invited to be interviewed during autumn 2019 to detail the impact of their involvement in the project on themselves, their pupils and their school, and to share any documentation relating to this impact. Interviews were conducted with 10 of the teachers (out of a possible 22) at various locations in England that were convenient for them. Each interview lasted between 30 and 60 min, was audio-recorded with the permission of the teachers involved, and was transcribed as soon as possible afterwards. Four of the ten interviewed teachers brought along documentation that they had produced in the form of pupil booklets and revised schemes of work.

The survey and interview data were analysed by means of the generation of themes using constructivist grounded theory methods [30]. A staged approach to the analysis was adopted, the first stage involved highlighting keywords and phrases within the responses. This involved colour coding frequently used terms within the transcriptions such as ‘surprising’, ‘solve’, ‘re-think’ and ‘reluctance’. The second stage focused on coding and grouping these into related issues, examples being participants’ reactions to the paradoxes and their varying experiences of using them to influence others. The third stage involved the identification of emerging themes such as participants wanting to address the paradoxes and having positive but frustrating experiences of using them to influence their colleagues and/or peers [31]. Constant comparison of new and emerging data was employed to explore how the data added to or changed the patterns of existing information. The trustworthiness of the findings was aided by asking participants to check the accuracy of their interview transcriptions.

All participants gave their informed consent to be involved prior to the commencement of this study. The research was conducted in accordance with the Declaration of Helsinki, and the protocol was approved by the Ethics Committee of the University in which the authors are employed (Project identification code C16-02 and C17-02).

## 3. Results

Analysis of the survey and interview data led to the emergence of three major themes associated with the PAL paradoxes, these being that participants (i) found the PAL paradoxes interesting, surprising and perplexing; (ii) expressed a keen desire to address and solve the PAL paradoxes; and (iii) experienced the joys and challenges of influencing colleagues’/peers’ health-related philosophies and pedagogies. The results are presented within these themes using illustrative quotations from a range of female and male trainee and experienced teachers across the different years of this study.

### 3.1. Finding the PAL Paradoxes Interesting, Surprising and Perplexing

All participants showed interest in the PAL paradoxes, as exemplified by the following quotations:


*I was fascinated by some of the contradictions. I hadn’t really thought much about this before but there it was in front of us, proper research to show they existed.*
(Female Trainee Teacher, 2016–2017)


*I found them really interesting. I’ve not read much research since my PGCE course but I found myself wanting to know more and more about them.*
(Male Teacher, 2017–2018, 3 years’ teaching experience)


*They made me sit up and think about what we’re doing in PE. I couldn’t wait to share them with my department, I knew they would also be interested in them.*
(Female Teacher, 2018–2019, 9 years’ teaching experience)

Many participants found the paradoxes surprising and perplexing, with some stating:


*I was really surprised by some of the paradoxes. Who would have thought that activity levels were so low in PE? It made me wonder if that’s the case with my own PE lessons. I couldn’t wait to find out.*
(Male Trainee Teacher, 2016–2017)


*I was blown away with them. In fact, I didn’t believe some of them to start with. But I looked up some of the papers….and there it was in black and white, so to speak.*
(Female Teacher, 2017–2018, 5 years’ teaching experience)


*I was a bit baffled and perplexed by some of them. Why are we getting things so badly wrong?*
(Female Teacher, 2018–2019, 4 years’ teaching experience)

### 3.2. Expressing a Keen Desire to Address and Solve the PAL Paradoxes

Most participants were keen to address the PAL paradoxes, as exemplified by the following quotations:


*I really don’t want to turn into a hypocrite, saying one thing and doing another. I need to make sure that what I believe in shows in my PE lessons.*
(Male Trainee Teacher, 2016–2017)


*We can’t live with these contradictions. We need to do something about them.*
(Female Teacher, 2017–2018, 6 years’ teaching experience)


*You can’t know all this and carry on regardless, can you? I think we need to go back to the drawing board and start again with our PE programme, I don’t think we’ve got the bigger picture right. This is going to change a lot of what we do.*
(Male Teacher, 2018–2019, 4 years’ teaching experience)

Indeed, many participants referred to some urgency about wanting to ‘sort out’ or ‘solve’ the contradictions posed by the paradoxes:


*If we know about all of this, why aren’t lecturers and researchers or the PE association urgently trying to solve these problems?*
(Male Trainee Teacher, 2016–2017)


*I decided that my department needed to do something about this straight away. We can’t be bleating on about wanting longer and more PE lessons if pupils are not very active in our PE lessons.*
(Female Teacher, 2017–2018, 2 years’ teaching experience)


*This all needs sorting out and quickly. It’s our responsibility to match what we say and do. It has made me do things differently for sure.*
(Female Teacher, 2018–2019, 8 years’ teaching experience)

The above quotations clearly demonstrate that the paradoxes influenced the health-related thinking and practices of many of the participants.

### 3.3. Experiencing the Joys and Challenges of Influencing Colleagues’/Peers’ Health-Related Philosophies and Pedagogies

In addition to the participants identifying changes to their own thinking and practices, as illustrated above, they also reported some success in influencing colleagues’/peers’ health-related thinking and practices. Examples of statements from both trainee and experienced teachers about this included:


*I talked to the other PE trainee at the same school about the paradoxes. He thought that all trainees should know about them, not just those who signed up for the project. We both timed activity levels in each other’s lessons and came up with ideas for increasing the levels… We also talked through each other’s fitness lesson plans and added in more links to pupils’ lifestyles, which aren’t in the department’s fitness unit. It was good to have someone to talk to about this.*
(Female Trainee Teacher, 2016–2017)


*I shared the paradoxes in departmental meetings, we focused on a different one each week. It made us all re-think what we’re doing and brought our thinking together more. We came up with a list of actions for the year and we reported back on them in our meetings…one action was to start the school year by telling the pupils about the ‘one hour a day’ recommendation and giving them an activity diary for the first half-term. We’ve also introduced personal activity programmes for the older pupils. Our programme is now much stronger and the pupils seem to like the changes we’ve made so we’re sticking with them.*
(Female Teacher, 2017–2018, 10 years’ teaching experience)


*I shared them at an academy event, they went down really well. Between us, we came up with loads of different ideas for solving some of the contradictions…the ideas included making sure that every unit of work includes a health message, helping pupils understand why fitness helps them in every day life, and sharing health and fitness apps that teachers have found useful for PE. I felt very good about the session afterwards…it prompted a lot of good debate and action. I know some colleagues are doing things differently in their schools as they are sending me emails to let me know how they’re getting on and I’m passing this information on to others.*
(Female Teacher, 2018–2019, 6 years’ teaching experience)

However, some participants also commented on frustrations and difficulties with attempting to change the thinking and practices of a minority of their peers/colleagues, as demonstrated in the following quotations:


*During my teaching practice…, I offered to talk about the PAL project in one of the PE department meetings and I showed the PE staff some of the paradoxes, the ones about low activity levels in PE and fitness testing putting some kids off activity. There was a lot of discussion but I’m not sure if it led to any changes, I doubt it as a few didn’t seem keen to change what they were doing.*
(Female Trainee Teacher, 2016–2017)


*Influencing some of my colleagues is going to take time, effort and energy. I don’t have the luxury of this at the moment. Everyone is so busy with other responsibilities…and tons of admin… It’s hard to do everything we’re supposed to do on the health front. I think I’ll just try things out in my own lessons first and then talk to colleagues about what worked well further down the line.*
(Female Teacher, 2017–2018, 2 years’ teaching experience)


*There’s a lot to do in your NQT (newly qualified teacher) year so I’ve not done as much as I’d hoped with my department this year. It’s not going to be easy, some of my colleagues are stuck in their ways, they don’t take kindly to newer members of staff, like myself, suggesting changes. I think you’ve got to prove yourself first, before they take much notice of you. One of them did ask for copies of the paradoxes though so there’s definitely some interest there.*
(Male Teacher, 2018–2019, 1 year’s teaching experience)

## 4. Discussion

It is evident from the findings that the paradoxes prompted various reactions and actions from the participants and that they had influenced their own and colleagues’/peers’ health-related philosophies and pedagogies. The fact that all participants found the PAL paradoxes interesting and most considered them surprising and even perplexing indicates that professional development which draws on a range of research-informed perspectives on health-related issues can engage teachers, serve to interrupt the status quo and lead to changes to teachers’ health-related thinking and practices. This is significant as previous research on teachers’ professional development in this area has cited both teacher engagement as well as teacher change (disturbing current thinking and practice) to be a particular but necessary challenge [32]. Indeed, that most participants expressed a keen desire to solve the PAL paradoxes suggests that they felt some degree of accountability or responsibility for the contradictions presented to them. It also indicates that the participants considered themselves empowered and capable of changing at least their own health-related thinking and practices. This serves to highlight the key role of teacher agency (that is, the potential to act in the interplay between personal capacities and contexts) in bringing about change, and the importance of reflection in strengthening teacher agency [33].

It was particularly encouraging that all participants experienced some degree of success in also influencing colleagues’/peers’ health-related philosophies and pedagogies. This experience also revealed some evolution of the use of the paradoxes over time in that, in order to influence others, the participants increased their awareness of the literature underpinning the paradoxes; and they became more confident in presenting examples of the paradoxes in action and offering practical solutions to address them. However, given what is known from literature on occupational socialisation in physical education, it is not surprising that some also experienced frustration and difficulties in doing so, especially those who were relatively early in their career and as yet held no or limited positions of responsibility within a department, such as trainee or newly qualified teachers [34,35]. The reported tension between pursuing education and health outcomes in schools [2] was furthermoreapparent in some participants’ responses with challenging workloads and limited time reportedly hindering their ability to influence colleagues’ health-related philosophies and pedagogies. Such pressures have similarly been cited to restrict the attention afforded to this area elsewhere [36].

Overall, the findings reveal that the use of PAL paradoxes incorporated within relevant professional development helped to change trainees’ and teachers’ health-related philosophies and pedagogies. This suggests that an explicit, albeit limited, focus on the intrapersonal and interpersonal levels of the social ecological model [22] through support for PE trainees and teachers (and, in turn, their colleagues and peers) can result in changes to young people’s experience of physical activity promotion within PE and the wider school setting. This is consistent with the findings of Weaver and colleagues [13,14] in the US and Bowler [15] and Sammon [16] in the UK, where improvements were detected in PE teachers’ effectiveness as promoters of physical activity following the implementation of new practices and a new pedagogical model respectively. This positive response was most likely associated with participants’ engagement in reflecting on and input into the PAL paradoxes, and with the sustainable features of the project such as its simplicity, accessibility and flexibility, all of which permitted its ongoing implementation across varying school settings and populations. Some of the changes to health-related philosophies and pedagogies identified by the trainees and teachers could be described as enduring and even transformative which suggests that the research-informed approach within the PAL project was successful in bringing about evidence-based teaching in the area of health-related physical education. We believe that this represents an authentic and constructive response to calls for evidence-based PE-for-health pedagogies [11,12].

This study has a number of strengths, in particular its close engagement with participants and the importance of the paradoxes being ‘real’ and credible in that they were based on trusted literature. This arguably led to many of the trainees and teachers remaining involved in the project throughout and to many feeling almost obliged to address the paradoxes. Additional key strengths include the relatively low-cost, low-burden approach of the research which also contributed to the good retention of the participants. A further strength was that it utilised the well-established socio-ecological model which has been effectively employed in previous studies of physical activity in school settings [21,22]. However, the project also has a number of limitations such as its relatively small and voluntary sample and measures not being included to verify the pedagogical changes reported by the participants (such as data from lesson observations or pupil focus groups). A further limitation is that, in contrast with previous studies which have included a focus on the organisational, community and policy levels of influence within the socio-ecological model [21,22], the PAL project did not address these higher structural levels which might have a more enduring health impact [22]. It is recommended that future studies expand the sample, include measures to verify the pedagogical changes and pupil responses (such as those adopted by Weaver et al. [13,14]), extend beyond the lower levels of the social ecological model, and further explore the influence of the communities of practice on pedagogical change.

## 5. Conclusions

This innovative low-cost, flexible and accessible approach to promoting active lifestyles can result in good teacher engagement in health-related professional development and in PE trainees and teachers transforming their health-related philosophies and adopting research-informed pedagogies, as well as influencing those of their colleagues/peers. This new approach therefore shows much promise in terms of developing sustainable, evidence-based teaching about leading active lifestyles, assisting PE teachers in their role as effective promoters of physical activity, and enhancing the subject’s contribution to public health. This is significant, given calls for new pedagogical approaches and teachers’ previously reported lack of engagement in professional development in this area.

## Figures and Tables

**Table 1 ijerph-17-07965-t001:** Summary of Study Participants from the Period 2015–2019.

Academic Year	New Participants	Continuing Participants	Total Number of Participants Who Engaged for the Academic Year
2015–2016	12 (3 teachers and 9 trainee teachers)	All participants were new	11 (2 teachers and 9 trainee teachers)
2016–2017	11 (2 teachers and 9 trainee teachers)	9 teachers from 2015–2016	19 (11 teachers and 8 trainee teachers)
2017–2018	9 (1 teacher and 8 trainee teachers)	9 teachers from 2015–2016 and 8 teachers from 2016–2017	25 (18 teachers and 7 trainee teachers)
2018–2019	No new participants were invited	8 teachers from 2015–2016, 7 from 2016–2017 and 7 from 2017–2018	22 teachers

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
