# Peer review of "Prompting Pedagogical Change through Promoting Active Lifestyles Paradoxes"

_ijerph, 2020, doi:10.3390/ijerph17217965_

Round 1
Reviewer 1 Report
An interesting paper, that addresses an important and relevant topic. This reviewer has a few minor questions/comments.
1) The authors mention in the Introduction (Line 52) that the PAL project was devised. The two papers that are referenced in Line 56 to support this are works from the same research group. However, while the first (Ref18, Line 349) highlights the principles for PAL (in a membership journal), the second (Ref19, Line 351) is currently in press. Whilst further details of the PAL project are addressed in the Materials and Methods, a brief introduction/description of the PAL project in the Introduction, would better place this current paradoxes paper into context for the reader.
2) Line 101 "During the professional development, ..." - does this "development" refer to the development workshop in the preceding sentence, or to the ongoing development of the participants?
3) Line 113 - Out of the four paradoxes highlighted, this one was the least clear to this reviewer/reader. A suggestion to clarify this for the reader would be to provide an example of each paradox.
4) Line 241 - "Male Teacher". The terms "trainee" and "teacher" are initially used in sections 1-3. In the abstract and section 4 (the Discussion), the terms "trainee" and "experienced" teacher are used. The quotation in lines 237-241 refer to a teacher completing an NQT year, and therefore raises the question of differentiation between trainee, newly qualified and experienced teachers (Line 266). Is this an artefact of the study whereby the participant joined the study as a trainee and has progressed, or is it a differentiation in the hierarchy that needs to be addressed that affects the study/participants perceptions?
5) The paper highlights the importance of addressing the paradoxes, but as a reader, this reviewer is left with the nagging questions of "So what are the examples?", "What were the interventions?". Whilst perhaps not within the remit of this paper and addressed, in part, by the authors in highlighting this as a limitation, this data would be most welcome.
6) Formatting issues - Lines 350 and 352
Author Response
1) Lines 55-60: A brief introduction/description of the PAL project has been added to the Introduction to contextualise this particular paper for the reader. I have tried to avoid unnecessary repetition of the information in the next section, and I've also included an example of a paradox to help clarify this early on in the paper.
2) Lines 104-105: It has now been made clear that the professional development in question does refer to the University-based professional development referred to in previous sentences.
3) Lines 59-60, 110-125, 132-135: Examples have been provided for each of the paradoxes referred to in the paper to ensure their clarity for the reader.
4) To ensure consistency and avoid confusion, the terms 'trainee' and 'teacher' have been used in the paper and the term 'experienced' teacher has been removed. Also, for each quotation, additional information about the participants has been added in terms of their teaching experience so the reader knows the stage they are at in terms of their teaching career.
5) Lines 229-231 and 235-237: I have looked back over the quotations to ensure that, collectively, they provide a range of examples of the changes that the participants made to their practices. To strengthen this particular aspect of the paper, I have added data demonstrating further examples of 'actions' and 'ideas' that are explicitly referred to in some quotations.
6) Lines 362-365: The formatting issues in these two references have been rectified.
Reviewer 2 Report
I appreciate the opportunity to review your manuscript. I think you’re focused on a key and important topic. What teachers believe about “Paradoxes”, think, do and used it on their practical classrooms is, certainly, a key quality practice in teaching. I do, however, have some suggestions for revision that I think would make your manuscript stronger.
If this paper focuses, as you said, on “PE trainee and experienced teachers’ engagement with the paradoxes”, and you had a longitudinal study, how that paradoxes evolves or not along the time and why, could be described in order to further enrich your work.
With regard to “pedagogies” finally only one model has been select, the social ecological model but no work about the use of that model in teacher training is referenced. Papers like
Hyndman, Brendon; Telford, Amanda; Finch, Caroline F.; and Benson, Amanda C. (2012) "Moving Physical Activity Beyond the School Classroom: A Social-ecological Insight for Teachers of the facilitators and barriers to students' non-curricular physical activity," Australian Journal of Teacher Education: Vol. 37: Iss. 2, Article 1. Available at: http://ro.ecu.edu.au/ajte/vol37/iss2/1
Langille J. L. D., Rodgers W. M. (2010). Exploring the influence of a social ecological model on school-based physical activity. Health Educ. Behav. 37 879–894. 10.1177/1090198110367877 - DOI - PubMed
Äli Leijen, Margus Pedaste & Liina Lepp (2020) Teacher agency following the ecological model: how it is achieved and how it could be strengthened by different types of reflection, British Journal of Educational Studies, 68:3, 295-310, DOI: 10.1080/00071005.2019.1672855
Could be used in orther to explain the paradoxes and the results.
Finally, regarding the description of the research, I think that the software used to treated the data is not mentioned, if one of them (like Atlas.it, QDA…) has been used. This part could be developed as well as the description of the participants is done, in order to undertand the perspective you used and how the categories and themes has been selected.
Again, I enjoyed reading your work. I hope you’ll find my suggestions helpful as you continue to work to share your research.
Author Response
1) Lines 274-277: I have considered this point and have added information to reflect how the use of the paradoxes evolved during the study.
2) Thank you very much for providing details of papers associated with the use of the social ecological model in promoting physical activity in school settings (Hyndman et al., 2012; Langille & Rodgers, 2010) and the ecological model of teacher agency (Leijen et al, 2020). I have used these to add depth to the paper.
I have also added these papers (and one other) to the references and re-ordered the reference list.
3) No particular software was used to treat the data but I have added more information to the data analysis part of the paper to further explain how the categories and themes were selected. I have also increased the description of the participants (in terms of their teaching experience) to permit further understanding of the range of perspectives of the trainees and teachers involved in the study.